# Strain-level metagenomic assignment and compositional estimation for long reads with MetaMaps

Alexander T. Dilthey [1,2], Chirag Jain[2,3], Sergey Koren[2] & Adam M. Phillippy[2]

Metagenomic sequence classification should be fast, accurate and information-rich. Emerging long-read sequencing technologies promise to improve the balance between these factors but most existing methods were designed for short reads. MetaMaps is a new method, specifically developed for long reads, capable of mapping a long-read metagenome to a comprehensive RefSeq database with >12,000 genomes in <16 GB or RAM on a laptop computer. Integrating approximate mapping with probabilistic scoring and EM-based estimation of sample composition, MetaMaps achieves >94% accuracy for species-level read assignment and $r^2 > 0.97$ for the estimation of sample composition on both simulated and real data when the sample genomes or close relatives are present in the classification database. To address novel species and genera, which are comparatively harder to predict, MetaMaps outputs mapping locations and qualities for all classified reads, enabling functional studies (e.g. gene presence/absence) and detection of incongruities between sample and reference genomes.

---

[1] Institute of Medical Microbiology and Hospital Hygiene, Heinrich-Heine-University Düsseldorf, Düsseldorf, North Rhine-Westphalia, Germany. [2] Genome Informatics Section, Computational and Statistical Genomics Branch, National Human Genome Research Institute, Bethesda, MD 20892, USA. [3] Georgia Institute of Technology, Atlanta, GA 30332, USA. Correspondence and requests for materials should be addressed to A.T.D. (email: alexander.dilthey@med. uni-duesseldorf.de)

Metagenomics, the study of microbial communities with the methods of genomics, has become an important tool for microbiology[1]. One key step in metagenomics is to determine the taxonomic entities that a metagenomic sequencing dataset is derived from. This can be done either at the level of individual reads (read assignment or taxonomic binning) or at the level of the complete dataset (compositional analysis or profiling).

A variety of methods have been developed for the analysis of metagenomic datasets, broadly falling into three classes. First, kmer-based read classifiers. This class includes approaches like Kraken[2], Kraken 2[3], Opal[4], CLARK[5], and MetaOthello[6]. Second, alignment-based methods, for complete genomes or signature or marker genes. This category includes tools like Megan[7,8], MetaPhlan[9], GASiC[10], and MG-RAST[11]. Third, Bayesian or EM-based estimators. This class includes Bracken[12], MetaKallisto[13], and Pathoscope[14,15]. There are also approaches based on linear models or linear mixed models, for example PhyloPythia[16,17], DiTASiC[18], and MetaPalette[19]; methods based on structured output support vector machines, for example PhyloPythia+[20]; methods that combine Markov models with kmers/alignment, for example Phymm/PhymmBL[21,22]; and methods that directly employ the Burrows-Wheeler transform[23], for example Centrifuge[24]. The large majority of these methods have been designed for the analysis of short-read data and only a small number of long-read-specific methods have been developed: Frank et al.[25] describe a method specifically developed for Pacific Biosciences CCS data, and MEGAN-LR[26] aligns long reads to protein databases and then carries out a lowest-common-ancestor-based analysis.

The dominance of short-read sequencing in the field of metagenomics has traditionally been driven by cost efficiency. However, long-read sequencing (defined here as reads >1000 bases) has recently become more cost-effective and has two intrinsic advantages over short-read sequencing for the interrogation of metagenomes. First, long reads preserve more long-range genomic information such as operon structures and gene-genome associations. The availability of this information can be key to functional and evolutionary studies, concerning, for example, the organization of metabolic pathways and horizontal gene transfer across metagenomes. Second, some long-read sequencers (the Oxford Nanopore MinION in particular) support rapid, portable and robust sequencing workflows, enabling "in-field" metagenomics. This is expanding the types of applications and scenarios that DNA sequencing and metagenomics can be applied to, such as the in-situ characterization of soil metagenomes in remote locations[27] or real-time pathogen sequencing during outbreaks[28]. For these reasons, the applicability and importance of long-read sequencing to metagenomics is growing rapidly.

This development of sequencing technology, however, has not yet been matched with the development of long-read-specific metagenomic analysis algorithms. Whereas tools that were designed for short reads can usually be applied to long-read sequencing datasets in principle, they often do not fully capitalize on the specific properties of the data. In short-read metagenomics, there are pronounced trade-offs between speed, accuracy and information richness. For example, methods like Kraken are very fast, but they do not attempt to determine the genomic positions of individual reads; alignment-based methods, on the other hand, can determine the genomic locations and alignment qualities of individual reads, but they are typically slow.

In the space of long-read metagenomics, desirable algorithms are both fast (to deal with large data volumes of incoming sequencing data on acceptable time scales, e.g., in the field)

and produce highly informative output that includes per-read positional and quality information (because the availability of long-range spatial information is one of the key advantages of long-read sequencing). Here we show that this is indeed possible by leveraging the specific properties of long reads in a new approach called MetaMaps.

MetaMaps implements a two-stage analysis procedure. First, a list of possible mapping locations for each long read is generated using a minimizer-based approximate mapping strategy[29]. Second, each mapping location is scored probabilistically using a model developed here, and total sample composition is estimated using the EM algorithm. This step also enables the disambiguation of alternative read mapping locations.

MetaMaps has three main advantages. First, utilizing a mapping approach enables MetaMaps to determine individual read mapping locations, estimated alignment identities, and mapping qualities. These can be used, for example, to determine the presence of individual genes, or to assess the evidence for the presence of novel strains or species (which will exhibit systematically decreased alignment identities). Second, our approach is robust against the presence of large "contaminant" genomes, introduced during sample collection and processing or part of the environmental DNA, which often lead to false-positive classifications in methods that rely purely on individual k-mers. Third, reliance on approximate mapping makes the algorithm much faster than alignment-based methods, and our mapping algorithm can be tuned to different read lengths and qualities.

MetaMaps is also well-equipped to handle the continuous growth of reference database size[30]. First, MetaMaps implements a "limited memory" mode that, while leading to slightly increased runtimes, reduces memory usage while maintaining the same level of accuracy. This enables, for example, complete mapping of a long-read metagenomic sample to a comprehensive NCBI RefSeq database on a laptop computer. Second, by using the EM algorithm for borrowing information across reads[13,15], MetaMaps can distinguish between closely related database genomes, a challenge that becomes more common as reference databases grow. The source package also includes support for Krona[31] and a set of lightweight R scripts for quick visualization of the sample-to-database mapping results.

Long-read mapping is challenging due to the high error rates of long-read sequencing platforms, such as Oxford Nanopore or Pacific Biosciences (PacBio). Even under the assumption of high error rates, however, significant numbers of short, exact, and approximately co-linear matches can be expected to connect a sequencing read to its correct mapping location if the read is long enough. For example, under a simple binomial model and the assumption of a uniform 15% error rate and a read length of 1000, there will be, on average, 73 exact 16-mer matches between a read and its correct mapping location. Searching for sets of consistently positioned short exact matches is therefore a promising long-read mapping strategy that is robust against sequencing error. Furthermore, it has been shown that the number of matches between a read and a putative mapping location can be used to approximate alignment identity[29]. Further improvements in terms of speed and memory consumption are possible by employing a minimizer-based[32] k-mer selection strategy tuned according to assumptions about read length and minimum alignment identity[29]. In this publication, our novel contributions are the development of a probabilistic mapping quality model; the incorporation of this model into an EM-based approach for the estimation of overall sample composition and composition-dependent mapping locations; and the integration of the core algorithmic components into a software suite to support applications in metagenomics.

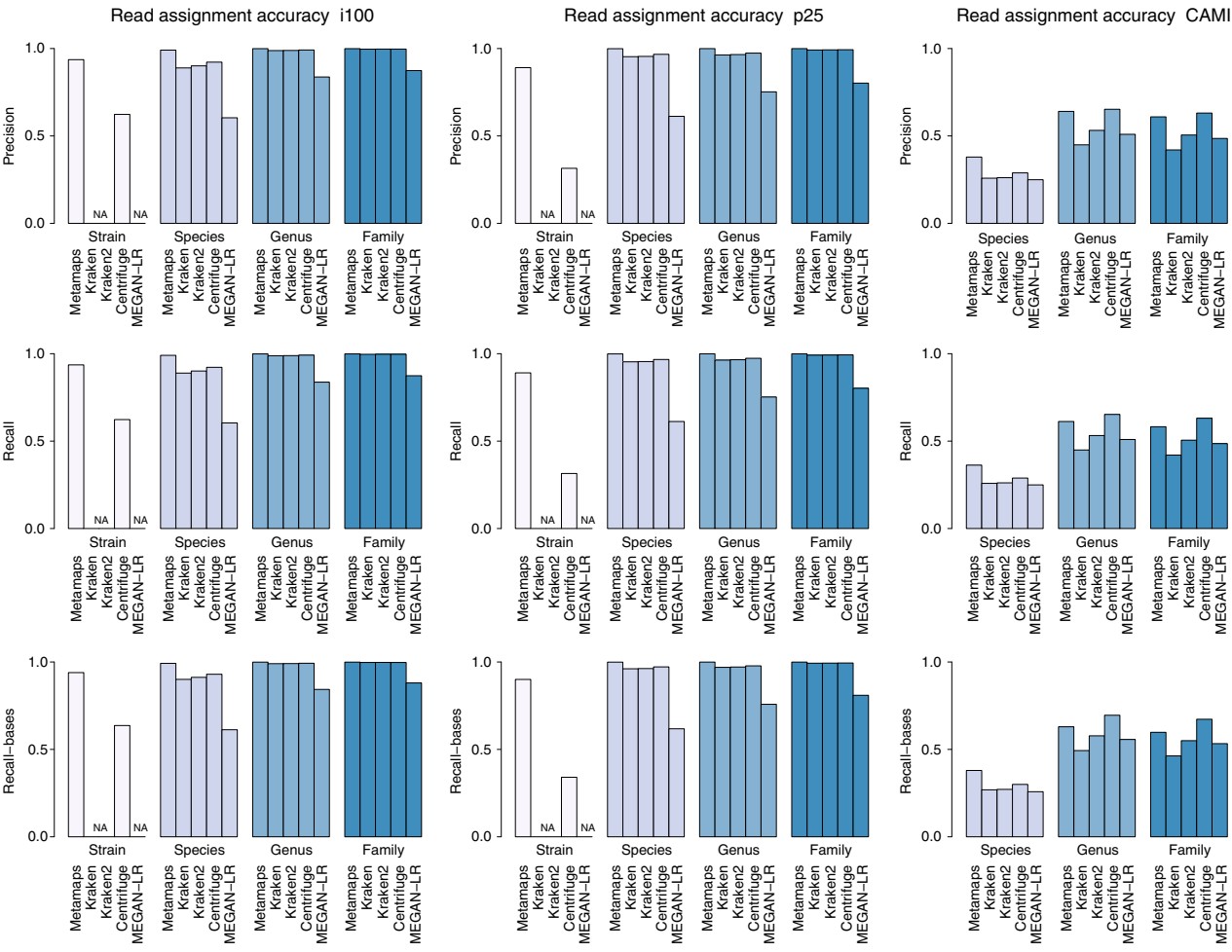

**Fig. 1** Read assignment accuracy in experiments with simulated data (i100, p25, and CAMI). Bar plots show precision, recall (read level), and recall (base level) for MetaMaps, Kraken, Kraken 2, Centrifuge and LAST + MEGAN-LR at different evaluation levels. Note that Kraken, Kraken 2 and MEGAN-LR were not designed to achieve strain-level resolution and therefore these tools are not evaluated at this level; also note that the CAMI experiment is not evaluated at the strain level because the CAMI truth set data are not consistently strain-level

## Results

**Performance on simulated data**. We first evaluate the performance of MetaMaps in two simulation experiments. Experiment i100 represents a medium-complexity metagenomic analysis scenario with approximately 100 species; experiment p25 a pathogenic metagenomic scenario with 15 strains of 5 potentially pathogenic bacteria and 10 other bacterial strains. At the strain level, MetaMaps assignments achieve a recall and precision of 94% (i100) and 89% (p25). At the species level, these metrics increase to ≥99%. MetaMaps typically outperforms Centrifuge by >30% in terms of recall and precision at the strain level. At the species level, MetaMaps outperforms Kraken, Kraken 2 and Centrifuge by 3–10% (precision) and 3–9% (recall). At higher levels, precision and recall of all tools that classify against a DNA database quickly approach or exceed 99%. The performance of LAST + MEGAN-LR remains lower and does not exceed 87% at the family level. Of note, accuracy of assignments to taxon IDs ≠ 0 (metric precision2) is high, reaching 99% at the family level; it remains, however, below that of MetaMaps. Per-read (Supplementary Data 1) and base-level (Supplementary Data 2) evaluation metrics are very similar for experiments i100 and p25. Read classification results are visualized in Fig. 1.

MetaMaps can also accurately estimate sample composition (Fig. 2). At the strain level, MetaMaps achieves a Pearson's $r^2$ between estimated and true abundances of 0.88 (i100) and 0.78

(p25). These increase to >0.99 at higher levels. The performance of Bracken and MetaMaps is similar; MetaMaps, however, exhibits slightly smaller distances (L1-norm) between estimated and true compositions. Centrifuge performs much worse than MetaMaps at the strain level and, at the species level, similarly to MetaMaps in experiment p25 and worse than MetaMaps in i100 (species-level $r^2 = 0.77$); its L1 distances to the true composition are elevated in comparison to MetaMaps and Bracken. For LAST + MEGAN-LR, $r^2$ remains below 0.66 across all levels and L1 is consistently higher than that of the other tools. Full compositional estimation accuracy results are shown in Supplementary Data 3.

We use the i100 experiment to assess the effect of read length on the ability to accurately classify a read. All methods show a trend towards higher classification accuracy for longer reads. For reads between 1000 and 10000 bases in length, this effect is most pronounced for Kraken (Fig. 3); the classification accuracy of MetaMaps is relatively constant for reads above the minimum length threshold. Importantly, for reads between 1000 and 5000 bases in length, the strain-level accuracy of MetaMaps is higher than or equivalent to the species-level accuracy of the other tools.

**Performance on real data**. To evaluate performance on real data, we apply MetaMaps to two sets of metagenomic sequencing data: PacBio RSII data from the Microbial Mock Community B of the

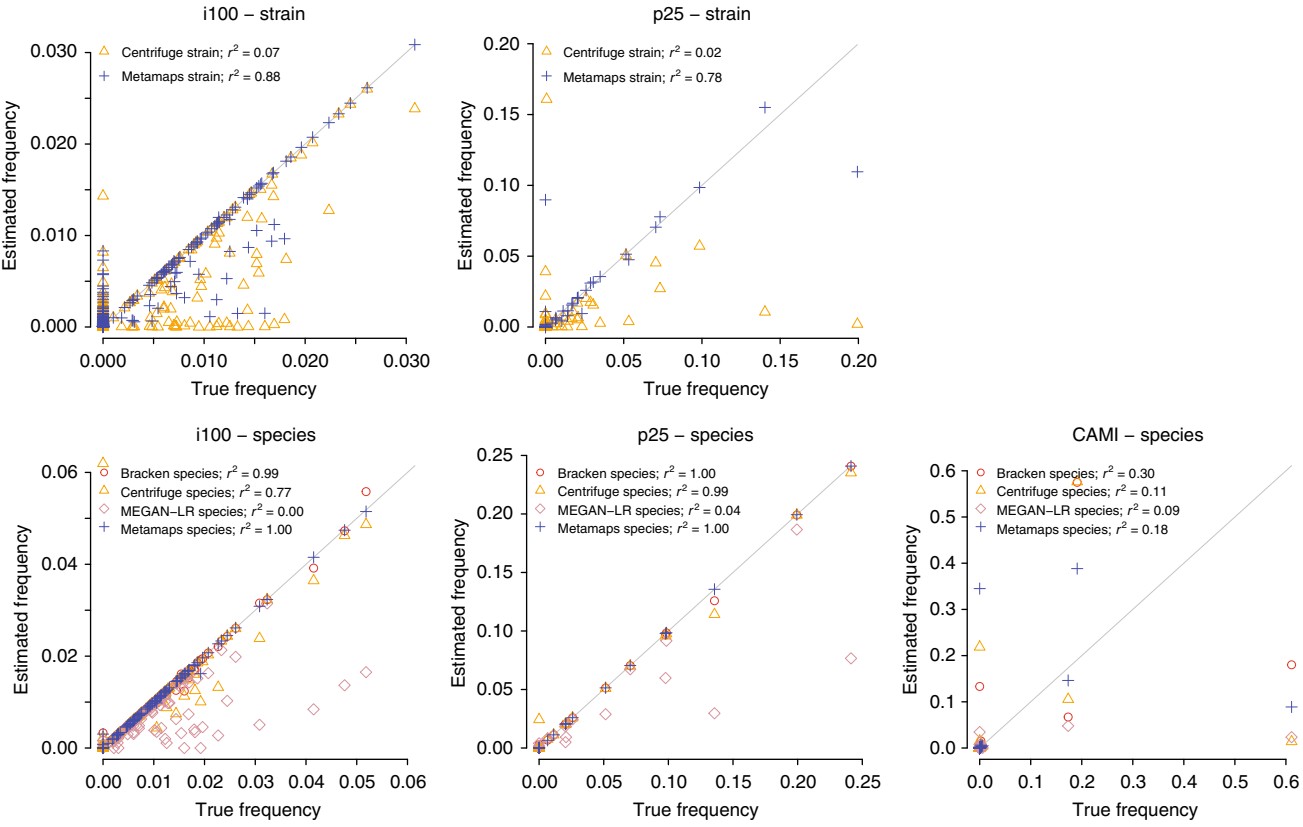

**Fig. 2** Inferred compositional estimates in the experiments with simulated data (i100, p25, CAMI). The figure shows the compositions inferred by MetaMaps, Centrifuge, Bracken and LAST + MEGAN-LR, compared to the true underlying composition. Note that two data points representing the "Unclassified" bin in the Centrifuge i100 and p25 strain-level experiments and two data points representing the "Unclassified" bin in the LAST + MEGAN-LR species-level experiments lie outside of the plotted area

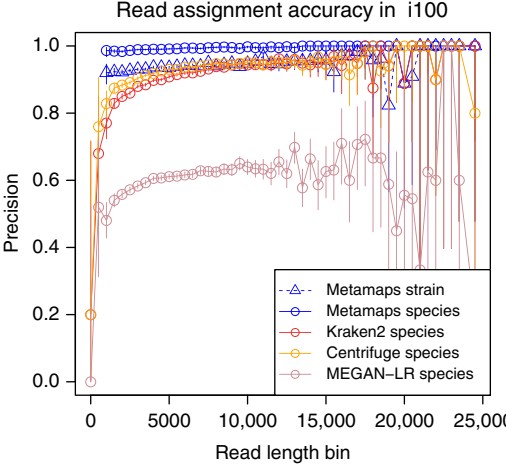

**Fig. 3** Precision of called reads in simulation experiment i100, stratified by read length. Note that MetaMaps results start at a minimum read length of 1000, corresponding to the "minimum read length" parameter the algorithm was run with. For bins above the MetaMaps minimum read length, precision equals recall. Kraken is outperformed by Kraken 2 and therefore not shown; Centrifuge stain-level accuracy is also omitted for clarity. Error bars represent 95% binomial proportion confidence intervals, computed using the normal approximation

Human Microbiome Project (HMP Set 7), referred to as "experiment HMP7"; and Nanopore GridION sequencing data of the Zymo Community Standards 2 synthetic community, referred to as "experiment Zymo".

First, we evaluate read assignment accuracy (Fig. 4 and Supplementary Data 1,2). Strain-level information is only available for the HMP experiment; MetaMaps achieves a precision of 65% and a recall of 61%, which increases to 64% at the base level. Consistent with observations on simulated data, the strain-level accuracy of Centrifuge is approximately 30% lower. We note that strain-level differences between the sequenced HMP7 sample and the reference genomes deposited in NCBI and as specified by HMP cannot be ruled out, which might contribute to the lower performance of MetaMaps on real as opposed to simulated data. Consistent with this, HMP7 precision increases to 95% (species) and 92% (genus, family) at higher taxonomic levels; Zymo precision is ~94% at all evaluated levels. Similarly, recall increases to 88% (species) and 86% (genus, family) in the HMP7 experiment, and to ~87% (all evaluated levels) in the Zymo experiment. When measured at the base level, precision remains approximately constant and recall increases by ~6% in both experiments.

When comparing MetaMaps to the other read assignment tools that classify against a DNA database, consistent patterns emerge. Across both experiments and all evaluated levels, the precision of the MetaMaps assignments is higher than that of the other tools. The magnitude of this effect depends on the taxonomic level (average difference of 5% at the species level, compared to 2% at the family level) and on the dataset (average difference across all levels for HMP7 of 5%, 2% for Zymo). On the other hand, MetaMaps typically achieves lower recall than the other tools; the average difference across the two datasets and all evaluation levels is 3%; the average difference in recall to the best-performing tool on HMP7 (Centrifuge) is 3%; to the best-performing tool on Zymo (Kraken 2), 6%.

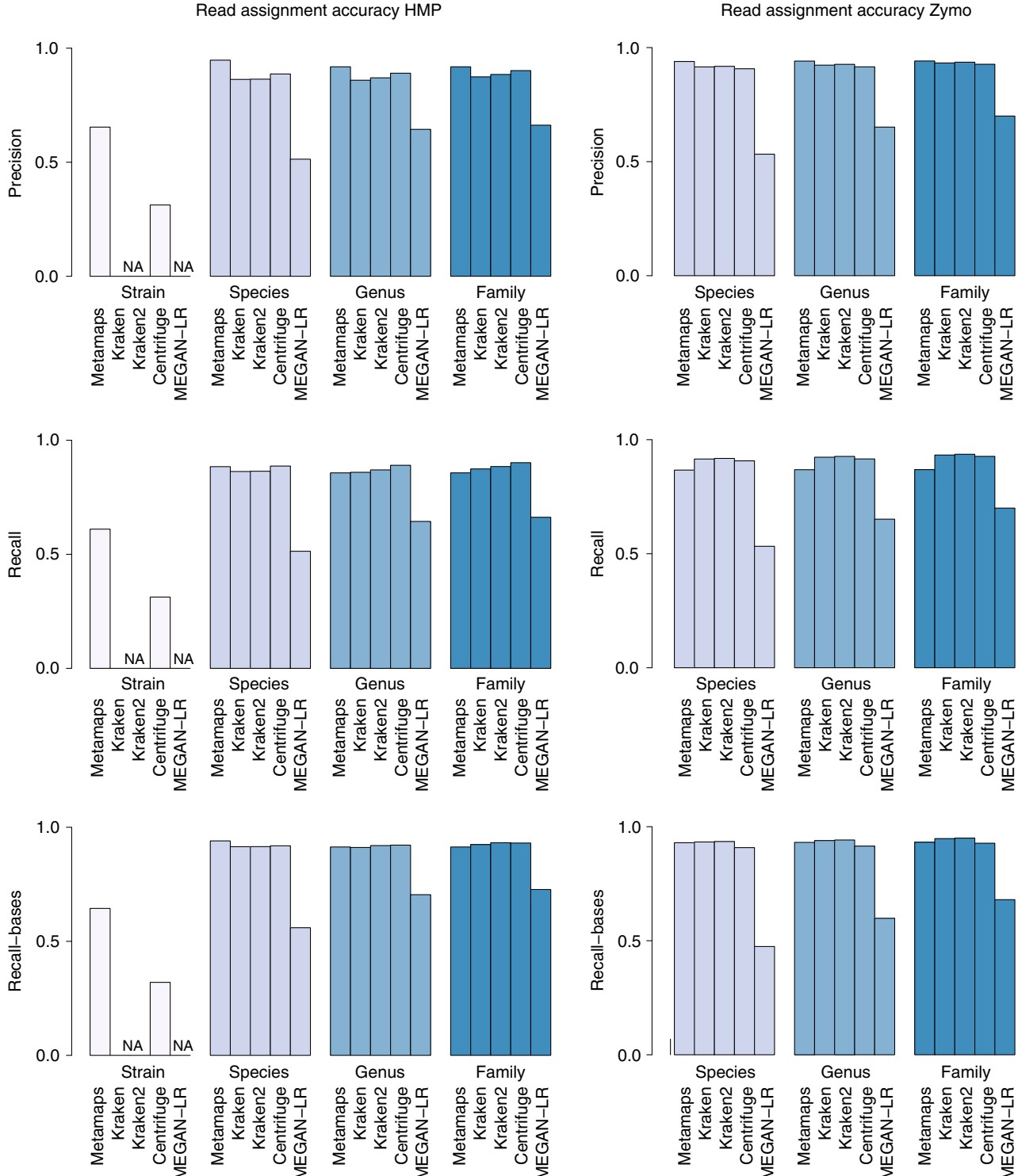

**Fig. 4** Read assignment accuracy in experiments with real PacBio (HMP) and Nanopore (Zymo) data. Bar plots show precision, recall (read level), and recall (base level) for MetaMaps, Kraken, Kraken 2, Centrifuge and LAST + MEGAN-LR at different evaluation levels. Note that Kraken, Kraken 2 and MEGAN-LR were not designed to achieve strain-level resolution and therefore these tools are not evaluated at this level

The lower recall of MetaMaps is explained by the fact that 7% (HMP7) and 8% (Zymo) of reads fall below the length threshold of 1000 bases. Although numerous by absolute count, these short reads make up a negligible fraction of the total bases sequenced, so when considering recall at the base level, the average difference in recall between MetaMaps and other tools is 0% (averaged over all tools and all levels). When considering only reads above 1000 bases in length, MetaMaps outperforms the other tools in terms of recall by 2% on average and in all individual evaluations, apart from the family level in experiment Zymo (where Kraken and Kraken 2 have an advantage of 0.1% and 0.3%, respectively).

The read-level assignment performance of LAST + MEGAN-LR, the only tool classifying against a protein database, is

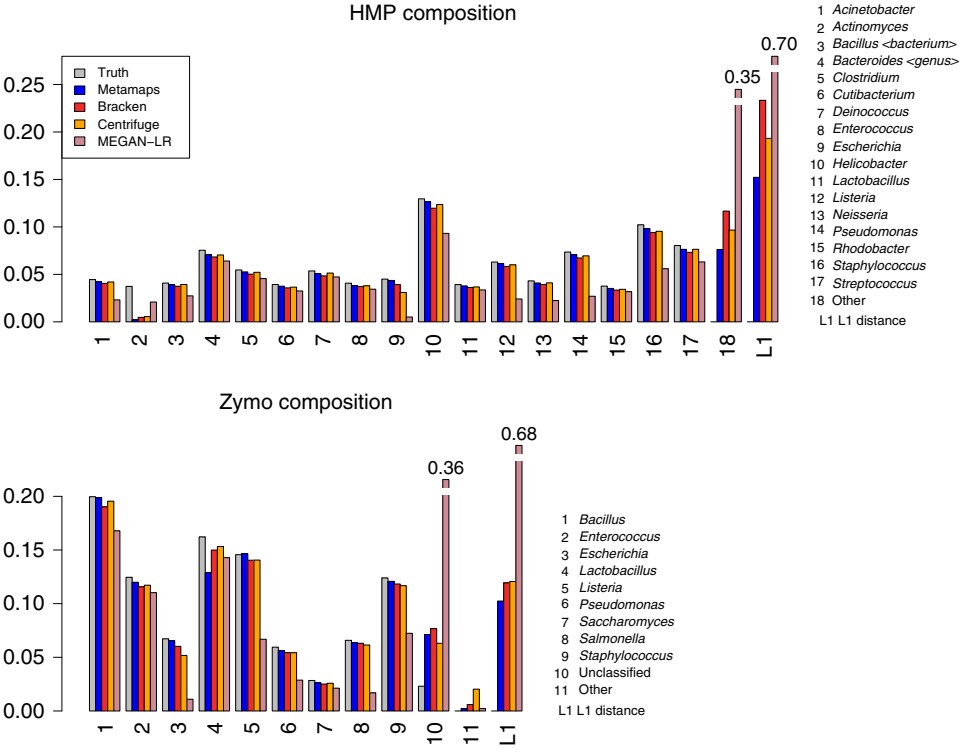

**Fig. 5** Compositional estimation in the HMP7 and Zymo experiment at the genus level. The figure shows the compositions inferred by MetaMaps, Centrifuge, Bracken, LAST + MEGAN-LR, and the assumed true composition. The L1 bar shows the combined L1 distance between the inferred and the true compositions. Note that the *Acinetobacter*, *Actinomyces*, and *Rhodobacter* genomes present in the HMP7 sequencing sample are not part of the database; reads classified as belonging to these genera map to other genomes of the same species/genus (see "Evaluating HMP7 performance" in Materials and methods). All but one fungal species (*Cryptococcus neoformans*) present in the Zymo sample are represented in the database

consistently lower than that of the other methods. Recall and precision range from 51 to 70%. Of note, and consistent with observations on simulated data, precision for reads assigned to taxon IDs ≠ 0 is high (≥95%; metric precision2), but consistently below that of MetaMaps.

Finally, it is interesting to note that almost all non-0 assignments (i.e., called reads not assigned to the 'unclassified' category) made by MetaMaps are correct; precision2 is close to or greater than 99% from the species level upward in both HMP7 and Zymo, and the precision2 of MetaMaps is higher than or equal to that of the other tools in almost all cases (50 of 51 comparisons across i100, p25, HMP7, and Zymo; Supplementary Data 1).

Second, we consider the accuracy of sample composition estimation (Supplementary Data 3 and Fig. 5). Consistent with read-level results, estimating sample composition at the strain level (only available for HMP7) is most challenging ($r^2 = 0.34$). The accuracy of compositional estimation is much higher at the species level ($r^2 = 0.98$ for HMP7 and 0.97 for Zymo) and at the genus/family levels ($r^2 = 0.90$ for HMP7 and 0.97/0.98 for Zymo). Of note, accuracy for the *Actinomyces* genus in HMP7 is low (Fig. 5) because the specific strain is not part of the reference database (see the section of the evaluation of HMP7 in "Methods").

We compare the accuracy of the compositional estimates of MetaMaps to these of Bracken, Centrifuge and LAST + MEGAN-LR. On HMP7, MetaMaps outperforms Bracken both in terms of $r^2$ (by a margin >0.1 at all levels) and L1. Centrifuge performance on HMP7 is broadly similar to that of MetaMaps, with slightly lower species-level $r^2$ (0.96) and slightly higher genus-/ family-level $r^2$ (0.91/0.93). In terms of L1, Centrifuge is less accurate than MetaMaps, though only by small margins

(0.02–0.08). LAST + MEGAN-LR produces less accurate sample composition estimates than the other tools on the HMP7 data, with $r^2 < 0.1$ and L1 consistently >0.5. On the Zymo dataset, compositional estimation generally exhibits high accuracy for all tools that classify against a DNA database ($r^2 \geq 0.96$ across all tools and all levels). MetaMaps outperforms Bracken by small margins (<0.01 for $r^2$ and <0.02 for L1). Compared to Centrifuge, the $r^2$ of MetaMaps is slightly lower (average difference 0.01) and its average L1 is lower (i.e., better) by a small margin (<0.02). Of note, though not specifically designed for compositional estimation, both Kraken and Kraken 2 produce accurate compositional estimates, with $r^2$ values ranging from 0.96–0.98 and low L1 distances, but they do not outperform MetaMaps. The compositional estimation accuracy of LAST + MEGAN-LR on the Zymo dataset is lower than that of the other methods and comparable to its performance on HMP7.

**Database-sample mismatches.** Incongruity between the sequencing sample and the utilized database (i.e., sequencing reads originating from strains or species not represented the database) is an important concern in metagenomics. To assess the behavior of MetaMaps on such datasets, we carry out three experiments.

First, we assess the effect of large out-of-database genomes— reflecting, for example, contamination with eukaryotic host DNA. Experiment e2 contains simulated reads from two eukaryotic genomes, neither of which is present in the reference database (the yellow fever mosquito and *Toxoplasma gondii*, representing plausible contamination scenarios). For both read sets, MetaMaps has a low false-positive rate and correctly leaves the large majority of reads unclassified (>99% precision/recall at the species, genus and family levels for mosquito and *Toxoplasma* reads); of note,

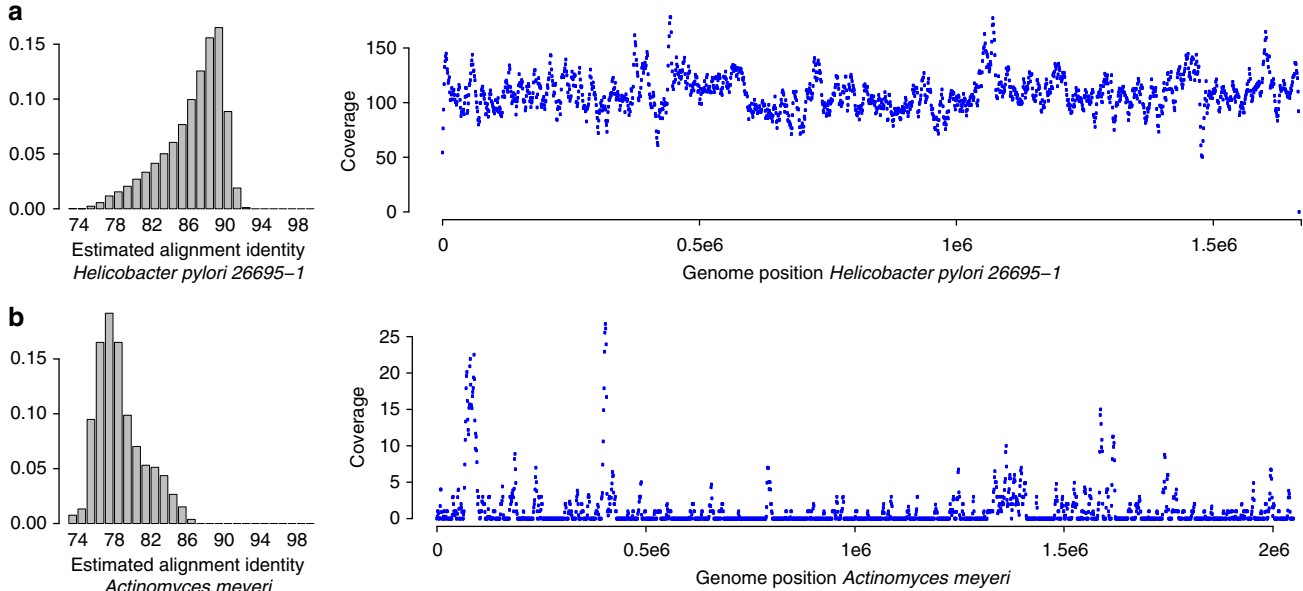

**Fig. 6** Alignment identity and spatial genome coverage for in- and out-of-database genomes. The figure shows estimated alignment identity and spatial genome coverage for *Helicobacter pylori 26695-1* (panel **a**) and *Actinomyces meyeri* (panel **b**) in the HMP7 experiment. *Actinomyces meyeri* is the next-closest database relative for the *Actinomyces odontolyticus* genome present in the HMP7 sequencing data (mash distance 0.14). *Helicobacter pylori 26695-1* is present in HMP7 and in the reference database. Uneven spatial coverage and estimated mapping identities shifted away from the expected mode around 0.88 for *Actinomyces meyeri* are indicative of a mismatch between the sequencing sample and the reference database. Complete plots for HMP7 are contained in Supplementary Data 4

the minimum length requirement of MetaMaps does not contribute to this result, as all simulated reads in this experiment are long enough. All other tools that classify against a DNA database have much higher mis-classification rates: that of Kraken varies between 17% (*Toxoplasma*) and 5% (mosquito); that of Kraken 2, between 10% (*Toxoplasma*) and 13% (mosquito); finally, that of Centrifuge is ~45% in both experiments (approximately half of these reads are classified as human, half as bacterial; a small minority is also counted as archaeal). The robustness of LAST + MEGAN-LR, which classifies against a protein database, against eukaryotic contamination is comparable to that of MetaMaps; precision and recall exceed or approach 98% for all evaluated levels in both experiments. Actual microbial contamination of these eukaryotic assemblies is possible, but given that MetaMaps shows similar sensitivity to the other tools on the other datasets this is unlikely to account for the observed misclassification rates.

Second, we use the HMP7 data to evaluate the effect of subtler mismatches and whether the availability of read mapping locations and estimated alignment identities enable the detection of database-sample mismatches. MetaMaps provides an R tool to visualize spatial coverage and identities of the read mappings. Examination of these plots for the *Actinomyces* genome (Fig. 6), which is diverged from the strain in the sequencing dataset (mash[33] distance 0.14), reveals both a highly uneven coverage pattern as well as a stark shift of read identities away from the expected average of around 0.88 (approximately equal to 1 minus the sequencing error rate, Supplementary Data 4). It is clear from these results that any *Actinomyces*-related result from this experiment would have to be interpreted with caution, consistent with the high evolutionary distance between the sample *Actinomyces* genome and its next-closest relative in the database.

Finally, in a third experiment referred to as "experiment CAMI", we investigate the effect of extensive sample-database incongruity for a complex sample. For this experiment, we use a simulated long-read dataset from the 2nd CAMI Challenge[34],

resembling a complex mouse gut metagenome. There are significant mismatches between the sample and the standard MetaMaps reference database; only 28% of species and 63% of genera present in the sample are represented in the database. The read-level classification accuracy of MetaMaps is mediocre; precision varies between 38% (species) and 64% (genus), recall between 36% (species) and 61% (genus). The other methods perform similarly (Fig. 1 and Supplementary Data 1); MetaMaps has a higher average precision than the best-performing alternative tool, Centrifuge (54% vs 52%), and almost identical recall (52%). Kraken, Kraken 2 and LAST + MEGAN-LR exhibit consistently lower read assignment accuracy than both Centrifuge and MetaMaps (precision and recall). Of note, however, the non-0 assignments of LAST + MEGAN-LR (metric precision2) are the most accurate of any evaluated method in the CAMI experiment (84% for LAST + MEGAN-LR; 78% for Kraken 2; 77% for MetaMaps, averaged over the evaluated levels).

Compositional estimation is similarly challenging; Centrifuge has the best average $r^2$ (0.57 vs 0.54 for MetaMaps and 0.33 for Bracken and LAST + MEGAN-LR; averaged over all levels), and MetaMaps exhibits the lowest average L1 distance (0.87 vs 0.92 for Centrifuge, 1.11 for Bracken, and 1.15 for LAST + MEGAN-LR). It is also worth noting that the there is a majority of false-positive calls (metric "binary precision") in the compositional estimates of all tools but LAST + MEGAN-LR; average binary precision is 3% for MetaMaps, 2% for Centrifuge, 33% for Bracken and 63% for LAST + MEGAN-LR (which also exhibits the highest average binary precision over all experiments and levels). While this observation is not unique to the CAMI dataset, it becomes particularly pertinent in combination with the overall low compositional estimation accuracy observed in this experiment.

These results clearly indicate that neither the compositional estimates produced by MetaMaps nor those produced by the other evaluated tools should be used to establish the definitive presence or absence of low-abundance taxonomic entities, in

particular for situations in which there is a significant incongruity between sample and reference database. Even LAST + MEGAN-LR, the method with the highest binary precision in the CAMI experiment, still produces many false-positive calls (49% at the species level and 28% at the genus level). Similar to what we observed for the *Actinomyces* case in experiment HMP7, however, the metrics produced by MetaMaps can be indicative of sample-database mismatches and of putatively unreliable calls. First, the median estimated MetaMaps alignment identity in the CAMI experiment is around 82%; this is suspiciously low for PacBio data (median alignment identity in the real HMP7 is around 88%, see Fig. 6), pointing towards database-sample mismatches. Second, binary precision can be significantly improved by removing low-identity entries from the compositional estimates. For example, removing all entries that have median alignment identities below 80% at the strain level (and their corresponding contributions at higher taxonomic levels) from the compositional estimate increases binary precision to 33% (species) and 47% (family), higher than that of any other of the evaluated tools but LAST + MEGAN-LR. Setting such a threshold, however, also reduces the accuracy of the overall estimate (average reduction in $r^2$ of 0.05), and it also reduces binary recall at the species level by 22%. Scripts for carrying out this filtering are part of the MetaMaps distribution.

**Runtime and memory-efficient mode**. A MetaMaps run in default mode requires 262 GB of RAM and between 16 and 210 CPU hours. Runtime depends on the size of the input dataset (ranging from 1 to 5 GB in the experiments presented here); on the content of the sequencing sample (analyzing experiment p25 takes approximately twice as much CPU time as analyzing experiment i100); and on the type of input data, with simulated data followed by PacBio and Nanopore data, possibly reflecting the more non-random error mode of the Nanopore technology. The resource requirements of MetaMaps are significantly higher than the corresponding requirements of Kraken/Bracken, Kraken 2, Centrifuge and LAST + MEGAN-LR (Supplementary Table 1). The runtime behavior and resource consumption of MetaMaps can be modified by three parameters. First, minimum read length: doubling the minimum read length to, e.g., 2000 bases halves memory consumption and can also reduce runtime by up to 50% (though not on all types of input data). The principal effect of an increased read length is a reduction in read-level recall (Supplementary Data 5). Second, at the expense of slightly increased runtimes, MetaMaps can be run in memory-efficient mode, with an upper memory consumption target specified by the user. We test this mode on both simulated and real data and find runtimes increased by a factor of 1.2–1.3 at a peak memory usage of 28 GB, and by a factor of 1.3–1.4 at a peak memory usage of 16 GB. Memory-efficient mapping thus enables the use of MetaMaps on a high-end laptop computer. The accuracy of both read assignment and sample composition is virtually unaffected by limiting memory (Supplementary Data 5). Third, we have implemented a multi-threaded version of MetaMaps; speedup relative to the number of utilized CPU cores is shown in Supplementary Fig. 1.

**COG assignment**. A unique feature of MetaMaps is the output of both mapping qualities and locations for all classified reads. To demonstrate the utility of read mapping locations, we carry out a COG (Clusters of Orthologous Genes[35]) analysis of the HMP7 data. Briefly, we map the HMP7 reads against a version of the MetaMaps database in which genes have been annotated with COG group assignments, and we count, for each COG group, how many read alignments overlap with genes annotated with this group (see Methods for details). The results of this analysis

are shown in Supplementary Fig. 2. COG category S (Function unknown) is followed by groups associated with basic bacterial metabolism and homeostasis, such as group K (Transcription) and group E (Amino acid transport and metabolism). Scripts for mapping read locations onto COG groups and other types of annotations output by eggNOG-mapper[36], such as Gene Ontology[37,38] terms, KEGG KOs[39] or BiGG metabolic reactions[40], are part of the MetaMaps distribution.

## Discussion

We have presented MetaMaps, an algorithm specifically developed for the analysis of long-read metagenomic datasets that enables simultaneous read assignment and sample composition analysis. The key novelty of MetaMaps is the combination of an approximate mapping algorithm with a model for mapping qualities and the application of the EM algorithm for estimation of overall sample composition. As discussed in the Introduction, this design was motivated by the aim to develop an algorithm tailored for long reads that is both fast and preserves per-read spatial and quality information.

Our evaluations show that MetaMaps produces the most accurate read assignments of all tools evaluated; in particular, whenever MetaMaps assigns a called read to a specific taxonomic entity, this assignment is correct in close to or larger than 99% of cases from the species level upward in the evaluated real datasets (metric precision2). Recall for long reads (above defined as >1000 bp) is consistently higher than that of the other tools, and recall at the base level (i.e., measuring how big a proportion of the generated experimental data is classified correctly) is very similar for MetaMaps and the evaluated methods.

Nevertheless, a proportion of reads remain unassigned under the MetaMaps model because they do not meet the minimum length requirement. This is a direct consequence of the approach we chose for approximate mapping, which determines minimizer density based on expected read lengths and alignment identities. Reads that fall below the chosen minimum length end up with minimizer sets that are too small to reliably determine their mapping locations. It is worth noting, however, that minimum read length and expected alignment identities are user-defined parameters that can be set empirically (for example based on the distribution of read lengths) and according to user preferences (e.g., with respect to runtime and the proportion of reads that remain unclassified). In addition, read lengths can be optimized with specific protocols for the extraction of high-molecular-weight DNA; the applicability of these, however, depends on sample and experimental conditions.

MetaMaps computes a maximum likelihood approximate mapping location, an estimated identity and mapping qualities for all candidate mapping locations. Its output is nearly as rich as that of alignment-based methods and enables a very similar set of applications, while being many times faster. There are multiple advantages to this approach. First, MetaMaps is robust against the presence of large out-of-database genomes, for example eukaryotic genomes. Contamination and environmental DNA are important concerns in many metagenomic studies, and the MetaMaps model is more robust against these than the other evaluated methods. Second, estimated alignment identities can be informative about the presence of spurious hits produced by novel species or strains in the sample which are not represented in the database. As we have shown in the HMP7 and CAMI experiments, visual analysis or automatic thresholding can contribute to detecting and dealing with database-sample mismatches. Scenarios with complex samples comprising many non-characterized entities, however, remain challenging; we discuss this point below. Third, because it reports mapping information,

MetaMaps can be used to ascertain the presence of particular genes or loci of interest, for example antibiotic resistance genes or virulence factors. We have demonstrated the feasibility of this by characterizing the abundance of COG groups in the HMP7 experiment.

In many plausible metagenomic analysis scenarios, computing resources are limited—for example when sequencing metagenomes using a portable nanopore device during a field trip without reliable internet connection. We therefore developed a feature to limit memory consumption during the approximate mapping step. As we showed, reducing memory consumption comes at a runtime cost, but accuracy remains unaffected, and, in contrast to many other similar approaches, classification is still carried out against the complete reference database.

Incongruity between the sequencing sample and the classification database, for example caused by the presence of yet-uncharacterized taxonomic entities, are an important concern in metagenomics. Microbial reference databases comprise but a fraction of total microbial genome diversity, and the likelihood of sample-database mismatches will depend on the source environment of the sequencing sample. In this publication, we explore the impact of database incongruity on the performance of MetaMaps. First, strain-level mismatches: these are explored in the Zymo experiment, representing a set of 10 genomes; of these, nine have no exact representation in the classification database (i.e., no strain-level match; Supplementary Data 6), but they are represented by species-level matches in the database. As we have shown, performance in this setting remains high. Second, individual species-level mismatches: these are explored in the HMP7 experiment at the example of *Actinomyces odontolyticus ATCC 17982*, the next-closest database relative of which belongs to a different species (*Actinomyces meyeri*) of the same genus. As we have shown, overall accuracy in this setting remains high, and the quality metrics produced by MetaMaps (estimated alignment identities and genome-wide coverage profiles) clearly pinpoint the mismatch. Third, widespread mismatches at the species and/or genus level, which are explored in the CAMI experiment. As we have shown, the accuracy of MetaMaps (and that of the other evaluated methods) decreases sharply in this setting, but the quality metrics of MetaMaps indicate the general presence of widespread database mismatches.

There are certain use cases in long-read metagenomics that are not well addressed by MetaMaps. In addition to the database-sample incongruities discussed above, this includes the analysis of samples with a high proportion of short reads. In these cases we recommend that users inspect the quality metrics produced by MetaMaps, in particular average estimated alignment identities and genomic coverage profiles, to judge the applicability of MetaMaps to a specific dataset. In some situations, alternative approaches, such as metagenomic de novo assembly[41–43], are likely to have fundamental advantages. Efforts like CAMI[34] will be essential to better understand the strengths, weaknesses and trade-offs associated with different algorithms and methods.

There are two important directions for the future development of MetaMaps. First, building support for streaming data into MetaMaps would be an important feature for many clinical applications. This could be used to control real-time sequencing experiments[44], in which, for example, the sequencing run is stopped when a certain genomic coverage has been reached, or when the estimate of sample composition has become sufficiently stable ("sequence until"). Such an extension would be relatively straightforward to implement in terms of the algorithms' architecture by dynamically re-computing mapping qualities, to the extent that they are influenced by changes in the global sample composition frequency vector. Second, it would be desirable to integrate an explicit term for genomic divergence directly into the statistical models of MetaMaps; this would enable the explicit detection of and testing for novel strains and species in the sequencing sample, the need for which was demonstrated in the CAMI experiment. k-mer painting approaches[19] have been suggested as a solution to this problem in the short-read space, but how to best implement the detection of novelty from long-read data remains an open question.

## Methods

**Strain-level metagenomic assignment**. "Strain-level" accuracy is technically defined here as source-genome resolution; that is, a read or a compositional estimate is counted as correct at strain-level resolution if and only if it is assigned to its correct genome of origin (known a priori for simulated data and determined via alignment for real data; see below). There are some instances in which a reference genome is directly attached to a species node in the NCBI taxonomy (typically when there is only one reference genome for the species); for these genomes, strain- and species-level resolution are synonymous.

**Reference database and strain-specific taxonomy**. A comprehensive reference database, comprising 12,058 complete RefSeq genomes and 25 gigabases of sequence, is used for all experiments presented here, unless otherwise stated. It includes 215 archaeal, 5774 bacterial, 6059 viral/viroidal, and ten eukaryotic genomes, seven of which are fungal and one of which is the human genome. The database also includes an extended, strain-specific version of the NCBI taxonomy, in which each input genome maps to exactly one node (see Supplementary Fig. 3). MetaMaps supports the use of custom databases, and scripts for downloading and processing genomes from NCBI are part of the distribution.

**Overview of the MetaMaps algorithm**. The goal of the MetaMaps algorithm is to estimate, for a sample of long-read metagenomic data, overall sample composition as well as the mapping locations of individual reads. Metagenomic mapping is complicated by the presence of closely related genomes in the reference database. Reads emanating from these will typically have high-scoring alignments on multiple reference contigs corresponding to different taxonomic entities. Under the assumption of a uniform prior over reference contigs, the placement of these reads remains ambiguous. In contrast to many classical mapping algorithms, MetaMaps therefore employs a composition-dependent prior. That is, the likelihood of each possible mapping location $i$ for read $r$ is modeled as $P_r(i) \times \frac{1}{E_{rg}} \times \mathbf{F}_g$, where $P_r(i)$ is the mapping quality of location $i$ for read $r$; $\mathbf{F}_g$ is the sample abundance of the corresponding taxonomic unit $g$; and $\frac{1}{E_{rg}}$ is a regularization factor. Conceptually, $P_r(i)$, the mapping quality, is similar to a normalized alignment score; it is independent of sample composition and quantifies the extent to which the sequence of the read agrees with the sequence of the potential mapping location. Mapping qualities are computed based on minimizer statistics. $\mathbf{F}$ is a vector that describes the composition of the sequencing sample; it is unknown a priori and iteratively estimated from the sample using the EM algorithm. In the following sections, we give a formal definition of the algorithm.

**Initial read mapping and identity estimation**. MetaMaps employs a fast approximate long-read mapping algorithm to generate an initial set of read mappings, fully described elsewhere[29] and visualized in Supplementary Fig. 4. Briefly, a minimizer[32] index is constructed from the reference database; intersections between the minimizer sets selected from a sequencing read and the reference define an initial set of candidate mapping locations. Low-identity candidate mapping locations are eliminated using a fast linear-time algorithm. For all surviving $N$ candidate mapping locations of a read $r$, alignment identity is estimated using a winnowed-minhash statistical model[29]. Briefly, let $S_r$ be the MinHash sketch of the read-selected minimizers; let $S_i$ be the MinHash sketch the set of reference-selected minimizers for mapping location $1 \leq i \leq N$; let $S_{r \cup i}$ be the MinHash sketch of the minimizer set union. We define $\widehat{J_i} := \frac{|S_{r \cup i} \cap S_i \cap S_r|}{|S_{r \cup i}|}$ as an estimate of the Jaccard similarity between the k-mer sets of the mapping location $i$ and the read, and further $\left( \frac{2 \widehat{J_i}}{1 + \widehat{J_i}} \right)^{\frac{1}{k}}$ as an estimator of the corresponding alignment identity for k-mer size $k$. Minimizer density is auto-tuned based on a user-defined minimum read length and minimum mapping identity (by default and for all experiments presented here, unless otherwise mentioned: 1000 bases and 80% identity).

**Mapping qualities**. Using the MinHash sketch, we define a probabilistic mapping quality model to quantify mapping uncertainty for reads with multiple mapping locations. Under the assumptions of the model and conditional on a known sequencing error rate $e$, we model $P(|S_{r \cup i} \cap S_i \cap S_r|)$ as binomial with parameters $n = |S_{r \cup i}|$ and k-mer survival rate $p = (1 - e)^k$. The posterior probability (mapping quality) of mapping location $i$ for read $r$ is defined as $P_r(i) = \frac{P(|S_{r \cup i} \cap S_i \cap S_r|)}{\sum_{j \in 1..N} P(|S_{r \cup j} \cap S_j \cap S_r|)} \cdot e$

is unknown and read-specific; for simplicity we define $(1 - e)$ as the estimated identity of the highest-scoring mapping for each read.

**Sample composition and read redistribution**. Let $G$ be the set of genomes in the database and let $F_g$ be the (unknown) probability that a sequencing read in the sample emanates from database genome $g \in G$. $\mathbf{F}$ is a vector that describes the sample composition and is to be estimated.

We define the likelihood of the mapped read set $R$ as $L_C(R; \mathbf{F}) := \prod_{r \in R} L_C(r; \mathbf{F})$ and the likelihood of an individual read $r$ as

$$L_C(r; \mathbf{F}) := \sum_{g \in G} \sum_{i \in \text{map}(r,g)} P_r(i) \times \frac{1}{E_{rg}} \times \mathbf{F}_g,$$

where $i$ is a read mapping location. To account for genomic duplications, $\text{map}(r, g)$ is the set of all mapping locations of read $r$ in genome $g$, $P_r(i)$ is the posterior probability of mapping location $i$, and $E_{rg}$ is the count of effective start positions for read $r$ in genome $g$. $E_{rg}$ is defined as contig length minus read length, summed over all contigs of genome $g$. This implies a uniform distribution over possible within-genome start positions of reads; for simplicity, we don't distinguish between circular and non-circular contigs.

We define the composition-dependent posterior probability of mapping location $i$ as

$$P_r(i; \mathbf{F}) = \frac{P_r(i) \times \frac{1}{E_{rg(r_i)}} \times \mathbf{F}_{g(r_i)}}{\sum_{g \in G} \sum_{j \in \text{map}(r,g)} P_r(j) \times \frac{1}{E_{rg}} \times \mathbf{F}_g}$$

where $g(r_i)$ is the genome associated with mapping location $i$ of read $r$. Summing over all reads, we obtain an updated sample composition estimate as

$$\mathbf{F}'_g = \frac{\sum_{r \in R} \sum_{i \in \text{map}(r,g)} P_r(i; \mathbf{F})}{|R|}$$

The frequency vector $\mathbf{F}$ is initialized with $\frac{1}{|G|}$ for each element and we update $\mathbf{F}$ until convergence of the likelihood $L_C(R; \mathbf{F})$. Unmapped reads exceeding the minimum read length are assigned to the 'unassigned' category (special taxon ID 0), followed by a final renormalization. Each mapped read $r$ is individually assigned to its maximum likelihood genome location.

**Memory-efficient mapping**. To enable classification against large reference databases with limited resources, MetaMaps supports the specification of a maximum memory target amount ("Memory-efficient mapping"). When run in memory-efficient mode, the order of contigs in the reference database is randomized and processed in a sequential manner. Starting from the first contig, index construction is started and continues until internally estimated memory consumption is just below the user-specified target amount or until the end of the reference database has been reached. The input data are then mapped against this index representing a subset of the reference database and stored on disk. The index is cleared, and construction of a new index begins at the position at which the process was previously aborted. Suboptimal mapping locations will later be assigned low mapping qualities during the EM step.

**Comparisons to other tools**. We compare MetaMaps to Kraken[2], Kraken 2[3], Bracken[12], Centrifuge[24], and MEGAN-LR[26]. Kraken is an archetypical tool for taxonomic assignment of reads; Kraken 2 is an updated and more memory-efficient version of Kraken. Bracken carries out sample composition estimation using a Bayesian approach. Centrifuge is the tool underlying Oxford Nanopore's "What's in my pot" (WIMP) application[45]. MEGAN-LR maps long reads against a protein database and uses a "lowest common ancestor" to place the reads in phylogeny. Kraken (and, by extension, Kraken 2) were not designed for compositional estimation; we therefore only consider these tools' performance at the level of individual read assignment in the main text and figures. For completeness, the Supplementary Data contain compositional information for these tools as well. Centrifuge and Megan are evaluated both for individual reads and sample composition. For each experiment, we use the same database for Kraken/Kraken 2, Bracken (with `--read-length = 2000`; note that this parameter is used to fit the read re-distribution model of Bracken and thus not comparable to the minimum read length parameter of MetaMaps), Centrifuge, and MetaMaps. For MEGAN-LR, we use an equivalent protein-level database, described below in a separate section. Kraken returns an explicit taxonomic assignment for each read and returns the special taxon ID 0 if a read remains unclassified. Bracken uses a Bayesian model to update and refine the read counts at lower levels of the taxonomy, so we call Bracken separately for re-estimation at the levels of species, genus, and family, and obtain a sample composition estimate by dividing the number of assigned reads per node by the number of total reads in the input dataset. After summing over all nodes, remaining reads are counted towards taxon ID 0. In cases in which Centrifuge returns multiple assignments for a single read, we assign the read to the lowest common ancestor (LCA) of the returned hits. Compositional estimates for Kraken, Kraken 2, Centrifuge and MEGAN-LR are computed from the individual read assignments and obtained by dividing the number of assigned reads per taxonomic entity by the number of total reads in the input dataset.

**Megan-LR database construction and sample analysis**. We use the LAST + MEGAN-LR workflow described in[26]. LAST[46,47] is used to align long reads in a frameshift-aware manner (parameters `-F15 -Q1`) against a protein reference database. The resulting output MAF files are sorted using MEGAN's `sort-last-maf` algorithm, converted to an RMA file using the `blast2rma` executable (parameters `-bm BlastX -lg -alg longReads`), and finally converted to taxonomic assignments using the `rma2info` tool (parameters `-r2c Taxonomy --ignoreUnassigned false`). To convert the (nucleotide) database used in all other experiments into a protein database, we use the NCBI Entrez service to obtain protein sequences for the genomes present in the nucleotide database. Specifically, we query the "nuccore" database for the accessions present in the nucleotide DB (e.g., NZ_CP008872.1), selecting the value "fasta_cds_aa" for the "rettype" parameter. We remove all protein sequences marked with the "pseudo" attribute, and substitute dash characters in the returned sequences with the character for the unknown amino acid ("X"). The complete protein set of the human genome is downloaded from the RefSeq FTP server and added to the sequences obtained via Entrez. The combined protein-level database has a size of approximately 11 gigabases and contains 21,138,770 entries. Long reads which don't generate a LAST alignment are treated as unclassified (taxon ID 0).

**Evaluation experiments**. We carry out six experiments to evaluate MetaMaps, covering multiple metagenome composition scenarios, simulated and real data (PacBio and Nanopore), sample contamination with eukaryotic DNA, and incongruity between the sequencing sample and the utilized database. For read simulation, we use pbsim[48] with parameters `--data-type CLR --length-mean 5000 --accuracy-mean 0.88`.

Experiment i100: The basis for this experiment is the "medium complexity" metagenome scenario described in[49], specifying a metagenome of 100 species with defined frequencies. The most recent records for the specified sequence accession numbers, all of which are present in the MetaMaps database, are obtained via the NCBI query interface, omitting 4 that were removed from RefSeq since publication of ref.[49] We use pbsim with the above parameters to simulate 1 gigabase of long-read data from the 96 genomes. The utilized accessions and the realized read counts are shown in Supplementary Data 7.

Experiment p25: The basis for this experiment are 25 bacterial genomes[14] comprising five strains of *Escherichia coli*, five strains of *Staphylococcus aureus*, five strains of *Streptococcus pneumoniae* and 10 other common bacterial strains, all of which are present in the MetaMaps reference database. We arbitrarily assign genome abundances according to a log-normal model and use pbsim to simulate 1 gigabase of long-read sequencing data. The utilized accessions and the realized read counts are listed in Supplementary Data 8.

Experiment e2: As a negative control, we use pbsim (with additional parameter `--length-min 2100` to ensure mappability of all simulated reads) to simulate long-read sequencing data from two eukaryotic genomes not present in the MetaMaps or Kraken databases. Specifically, we simulate 1 gigabase of sequencing data from the *Aedes aegypti* (yellow fever mosquito) genome (GCF_002204515.2), and 1 gigabase of sequencing data from the *Toxoplasma gondii* ME49 genome (GCF_000006565.2). The two read sets are analyzed independently with MetaMaps and the other tools.

Experiment HMP7: For the HMP7 experiment, we use the "Set 7" data (2.9 Gb FASTQ; 319,703 reads) of the PacBio HMP sequencing experiment, based on genomic DNA (Even, High Concentration) from Microbial Mock Community B. To generate a truth set, we use bwa mem[50] with `-x pacbio` to map the reads against the reference genomes specified for the DNA source. All reads that cannot be mapped with bwa are excluded, and the primary alignment for each read determines the assumed true placement. The accession numbers of the reference genomes and their realized read counts are listed in Supplementary Data 9.

Experiment Zymo: For the Zymo experiment, we use approximately 5 Gb of randomly sampled reads from an Oxford Nanopore GridION sequencing run[51] on the Zymo Community Standards 2 (Even) mock community (Batch ZRC190633). The Zymo Community Standards community comprises five Gram-positive bacteria, three Gram-negative bacteria, and two types of yeast. The generated read data are publicly available (https://github.com/LomanLab/mockcommunity). The Zymo Specification Sheet provides a species-, but not a strain-level description of the community; therefore, no strain-level evaluation is carried out for the Zymo set. To verify that none of the Zymo genomes (with the possible exception of the *Bacillus subtilis* genome) have a strain-level match in the MetaMaps database, we compute the minimum mash[33] distance between the Zymo-provided reference genomes to the closest in-database genome. To generate a read-level truth set, we use BWA-MEM[50] with `-x ont2d` to map the reads against the reference genomes provided by Zymo. All reads that cannot be mapped with bwa are excluded, and the primary alignment for each read determines the assumed true placement. The 10 species, their realized read counts and the computed mash distances are listed in Supplementary Data 6. Note that one of the fungal species present in the Zymo sample, *Cryptococcus neoformans*, is not part of the utilized database.

**Experiment CAMI**: For the CAMI experiment, we use a simulated long-read dataset from the 2nd CAMI Challenge[34], representing a PacBio-sequenced metagenomic community of a mouse gut ("2nd CAMI Challenge Mouse Gut Toy Dataset"). The CAMI dataset comprises approximately 5 Gb of sequencing data and comes with a read-level truth dataset. No strain-level evaluation is carried out, as not all reads are assigned to a strain-level node in the truth dataset.

**Read-level evaluation metrics**. Read assignment accuracy is assessed by precision (the proportion of correct read assignments; also referred to as positive predictive value or 'PPV') and recall (the proportion of reads having been given a correct assignment); see below for mathematical definitions. Specifically, there is a true taxonomic assignment $\text{truth}(r) \in T$ for all reads $r$ in the validation set $V$ and an inferred taxonomic assignment $\text{inference}(r) \in \{0 \cup T\}$ for some or all reads $r$ in the validation set, where $T$ is the set of nodes in the database taxonomy and 0 is a special taxon ID indicating unassignedness. The function $\text{to\_level}(a,l)$, where $a \in \{0 \cup T\}$ is a taxonomic assignment and $l$ a taxonomic level, enables the conversion of these assignments to specific taxonomic levels and is defined as

$$\text{to} - \text{level}(a,l) := \begin{cases} a, & \text{if level}(a) = l \\ \text{ancestor}(a,l), & \text{if level}(a) < l \\ 0, & \text{if level}(a) > l \\ 0, & \text{if } a = 0 \end{cases}$$

where $\text{level}(a)$ is a function that returns the taxonomic level of an assignment $a$ and where $\text{ancestor}(a,l)$ is a function that returns the $l$-level ancestral node of $a$. An assignment is counted as correct at level $l$ iff $\text{to\_level}(\text{inference}(r), l) = \text{to\_level}(\text{truth}(r), l)$. For the evaluation of strain-level correctness, we define that $\text{to\_level}(a,\text{'strain'})$ be $a$ iff there is a mappable genome attached to $a$ in the classification database, and 0 otherwise. The value of the function $\text{correct}(r, l)$ is defined as 1 if a read assignment is correct at level $l$ and 0 otherwise. Precision at level $l$ is then defined as $\sum_{r \in V'} \text{correct}(r,l)/|V'|$, where $V' \subseteq V$ is the set of reads in the validation set that have an assignment; and recall at level $l$ is defined as $\sum_{r \in V'} \text{correct}(r,l)/|V|$. MetaMaps produces assignments for all reads in the validation set longer than 1000 bp, some of which might be 0. Note that the function to\_level will convert non-leaf taxonomic assignments to 0 when evaluating at lower taxonomic levels.

Note that the presence of 0s is not confined to the inference set; $\text{to\_level}(\text{truth}(r), l)$ is 0 if the taxonomic level of $\text{truth}(r)$ is higher than $l$. This can happen in experiments e2, HMP7, Zymo, and CAMI for reads that emanate from out-of-database genomes; for these, $\text{truth}(r)$ is set to the node that represents the most recent common ancestor between the source and database genomes (see below and Supplementary Fig. 5).

Reads carrying with an assignment $a$ in the inference set for which $\text{to\_level}(a, l) = 0$ at level $l$ could also be considered as entirely uncalled at $l$; we therefore also report the metric precision2, which, for level $l$, is defined as the proportion of correct non-0 assignments at level $l$. That is, precision2 is defined as $\sum_{r \in V''(l)} \text{correct}(r, l)/|V''(l)|$, where $V''(l)$ is the set of reads $r \in V$ for which an assignment has been made and for which $\text{to\_level}(\text{inference}(r), l)$ is not equal to 0.

**Base-level evaluation metrics**. In addition to the number of correctly quantified reads, the number of correctly classified bases (i.e., the proportion of experimental data generated) is also a relevant metric. We therefore also report precision-bases, precision2-bases and recall-bases, which correspond to read-length-reweighted versions of precision, precision2 and recall.

**Compositional evaluation metrics**. For the evaluation of sample composition estimation accuracy, the true and inferred sample composition vectors (always scaled to the complete set of reads, see above) are transformed to the desired level using the function to\_level defined above. The accuracy of the inferred composition is then assessed by the two metrics L1 (the distance between the true and inferred sample composition vectors using the L1 norm) and $r^2$ (Pearson's $r^2$ for the true and inferred composition vectors); both composition metrics are computed over columns that have value >0 in either vector. In addition, we also report recall and precision on the presence and absence of taxonomic entities at the compositional level ("binary classification metrics",[34]). Binary recall and binary precision at the compositional level measure how well the compositional estimates capture the presence or absence of taxonomic entities, independent of the assigned abundance.

**Evaluating HMP7 performance**. HMP7 comprises 20 microbial strains. Of note, 3 of these are not part of the MetaMaps database, because the corresponding assembly records as specified in the Mock Community B Product Information Sheet have been removed from RefSeq or are not classified as "complete genome". For two missing genomes (*Acinetobacter baumannii ATCC 17978* and *Rhodobacter sphaeroides 2.4.1*), the MetaMaps database contains closely related genomes of the same species (mash[33] distances 0.00 and 0.01, respectively). For the remaining genome (*Actinomyces odontolyticus ATCC 17982*), the next-closest in-database relative belongs to a different species (*Actinomyces meyeri*, mash distance 0.14). For the truth set that read assignments are evaluated against, out-of-database genomes

are assigned to non-leaf nodes of the taxonomy (see above); specifically, the true strain-level taxon IDs for all *Acinetobacter* and *Rhodobacter* reads, and the true strain- and species-level taxon IDs for all *Actinomyces* reads, are set to 0 (Supplementary Fig. 5). Note that MetaMaps will always assign all mapped reads at the strain level, whereas Kraken and the other evaluated tools can place reads at higher taxonomic levels.

**COG assignment**. To illustrate the utility of read mapping positions in a metagenomic setting, we map the HMP7 data against an extended version of the MetaMaps database in which genes have been annotated with Clusters of Orthologous Groups (COG[35]) information. The extended version of the database comprises 24,945 genomes from RefSeq; in addition to the DNA sequences of the genomes, we also obtain the positions and amino acid sequences of the encoded genes. As part of the database construction process, we use eggNOG-mapper[36,52] to obtain a COG assignment for each protein-coding gene. For each gene and its associated COG assignment, we aggregate the number of overlapping mapped HMP7 reads.

**Reporting summary**. Further information on research design is available in the Nature Research Reporting Summary linked to this article.

## Data availability
The HMP7 data are publicly available (https://github.com/PacificBiosciences/DevNet/wiki/Human_Microbiome_Project_MockB_Shotgun). The Zymo data are publicly available (https://github.com/LomanLab/mockcommunity). The CAMI data are publicly available from the CAMI website (https://data.cami-challenge.org/participate; "2nd CAMI Challenge Mouse Gut Toy Dataset"; "19122017_mousegut_pacbio_scaffolds/2018.02.13_14.02.01_sample_0"). The simulated sequencing reads and the utilized databases have been archived at OSF (https://doi.org/10.17605/OSF.IO/XY4VN). All other relevant data is available upon request.

## Code availability
MetaMaps and all associated code are available on GitHub (https://github.com/DiltheyLab/MetaMaps).

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

## Acknowledgements

This work has been supported by the Intramural Research Program of the National Human Genome Research Institute, National Institutes of Health, and by the Jürgen Manchot Foundation.

## Author contributions

A.T.D. and A.M.P. conceived the study; A.T.D., C.J., S.K., and A.M.P. designed the method; A.T.D., C.J., and S.K. analyzed data; and A.T.D., C.J., S.K., and A.M.P. wrote the paper.

## Additional information

**Competing interests:** The authors declare no competing interests.

