## [Peer Review File · Nature Communications]

Editorial Note: This manuscript has been previously reviewed at another journal that is not operating a transparent peer review scheme. This document only contains reviewer comments and rebuttal letters for versions considered at Nature Communications .

Reviewers' comments:

Reviewer #4 (Remarks to the Author):

I did not review the original submission of this manuscript but it seems to me that the authors have addressed the original set of responses raised by the reviewers. I found the revised manuscript to be a detailed but easy to follow description of what could be a rather useful algorithm. There are going to be increasing amounts of long read metagenome data generated in the near future and tools such as this that can classify long reads efficiently are much needed. However, I do still have some concerns regarding this manuscript that should be addressed:

1) The revised manuscript now includes a detailed description of software for metagenome taxonomic classification. However, their description of MEGAN-LR is incorrect. It is not limited to protein analysis. It matches nucleotide reads against protein databases (the NCBI-NR). It can definitely be used for long read taxonomic classification as is detailed extensively in the manuscript. The authors should correct this and add a comparison between their software and MEGAN-LR. Since it uses an LCA algorithm it may be more robust to the situation where the community comprises mostly novel genomes. Therefore, a comparison on the CAMI challenge data would be particularly relevant.

2) Again, regarding the sensitivity of this tool to genomes that are missing from the database. For most real-world applications the majority of organisms will lack sequenced genomes. Even for the human gut if we sequence any given sample more than half of the strains will be novel in my estimation. This has been confirmed by recent large-scale binning studies (Pasolli et al. Cell 2018). The authors have discussed this in the manuscript but I still think that their abstract, which may be as far as some readers get, will give a false sense of the real-world accuracy which will not be 94% at the species level. A corollary needs to be added that this applies when the community genomes are in the Database.

3) I was very keen to try this tool myself. I took some Nanopore reads recently generated from an AD reactor. The 'mapDirectly' step worked fine but the 'classify' step threw the following error:

```
>>>>>>>>>>>>>>>>>>>>>>
DB = databases/miniSeq+H/
Mappings = classification_results
Min. reads for 'U' = 10000
Threads = 1
>>>>>>>>>>>>>>>>>>>>>>
Read taxonomy from databases/miniSeq+H//taxonomy -- have 17771 nodes.
Starting EM...
EM round 0
terminate called after throwing an instance of 'std::out_of_range'
what(): stod
Aborted (core dumped)
```

The AD community will be rather novel so this could be related to a lack of genomes in the Database as above. It would be nice if the authors have any insight as to the above error?

We would like to thank the reviewer for raising these important points!

We now submit an improved version of the paper, addressing the concerns raised by the reviewer and incorporating all suggestions. A point-by-point response is enclosed below, with reviewer comments in blue and our responses in black. We summarize the most important changes here:

- We have modified the abstract to make clear that the stated accuracies refer to situations in which there are no large database-sample incongruities.
- We have corrected the description of MEGAN-LR and added it to all validation experiments.

1) The revised manuscript now includes a detailed description of software for metagenome taxonomic classification. However, their description of MEGAN-LR is incorrect. It is not limited to protein analysis. It matches nucleotide reads against protein databases (the NCBI-NR). It can definitely be used for long read taxonomic classification as is detailed extensively in the manuscript. The authors should correct this and add a comparison between their software and MEGAN-LR. Since it uses an LCA algorithm it may be more robust to the situation where the community comprises mostly novel genomes. Therefore, a comparison on the CAMI challenge data would be particularly relevant.

We thank the reviewer for pointing out that our description of MEGAN-LR was incorrect! We have clarified the description in the introduction and now make clear that MEGAN-LR can indeed be used for the classification of long (DNA) reads.

As suggested, we have also added MEGAN-LR to all validation experiments.

We find that the read classification accuracy of MEGAN-LR is generally lower than that of the other tools – for example, precision remains at or below 87% on simulated data (experiments i100, p25; v/s 99% for MetaMaps), and it does not exceed 70% on real data (experiments HMP7, Zymo; v/s \geq 92% for MetaMaps).

As predicted by the reviewer, however, there are some metrics particularly in the CAMI experiment on which MEGAN-LR performs significantly better than the other tools – most importantly “binary precision”, an abundance-independent metric that measures the proportion of predicted taxonomic entities present in the sample. Here, Megan-LR results range from 51% (species) - 72% (genus), which is approximately twice as good as the next-best performing tool (Centrifuge, 22% at ‘species’ and 38% at ‘genus’).

At corresponding false-positive rates of 28% - 49%, accurately inferring the presence or absence of specific taxonomic entities from long-read metagenomic data remains challenging, even when using MEGAN-LR. However, we now state explicitly in the discussion of the CAMI results that MEGAN-LR achieves the highest performance for ‘binary precision’.

2) Again, regarding the sensitivity of this tool to genomes that are missing from the database. For most real-world applications the majority of organisms will lack sequenced genomes. Even for the human gut if we sequence any given sample more than half of the strains will be novel in my estimation. This has been confirmed by recent large-scale binning studies (Pasolli et al. Cell 2018). The authors have discussed this in the manuscript but I still think that their abstract, which may be as far as some readers get, will give a false sense of the real-world accuracy which will not be 94% at the species level. A corollary needs to be added that this applies when the community genomes are in the Database.

REVIEWERS' COMMENTS:

Reviewer #4 (Remarks to the Author):

I appreciate the efforts that the authors have made to address my comments. I still cannot get the AD reads to run even after updating from the github but I will create a github issue regarding that. I am confident that it is simply due to the low number of hits encountered in this data set. I am happy for this very interesting study to now be published and look forward to reading the final version of the paper.

Chris Quince

Response to reviewers

I appreciate the efforts that the authors have made to address my comments. I still cannot get the AD reads to run even after updating from the github but I will create a github issue regarding that. I am confident that it is simply due to the low number of hits encountered in this data set. I am happy for this very interesting study to now be published and look forward to reading the final version of the paper.

Chris Quince

We would like to thank Prof. Quince for reviewing our manuscript and for the suggested changes, which have greatly strengthened our work!

We have contacted Prof. Quince directly and we are currently in the process of debugging this minor software issue, which, as noted by Prof. Quince, is likely related to the low number of encountered hits.